# Effect of EphA2 Silencing on Inhibiting the Progression of Renal Cell Carcinoma in an Orthotopic Mouse Model

**DOI:** 10.3390/cells14241981

**Published:** 2025-12-13

**Authors:** Taein Lee, Hye-Sun Lee, Sangjun Yoo, Hoyoung Bae, Min Chul Cho, Junghoon Lee, Hyeon Jeong

**Affiliations:** 1Department of Urology, Seoul Metropolitan Government Boramae Medical Center, Seoul National University College of Medicine, Seoul 07061, Republic of Korea; xodlsrrr@snu.ac.kr (T.L.);; 2Department of Urology, Seoul National University Hospital, Seoul 03080, Republic of Korea

**Keywords:** EphA2, renal cell carcinoma, Renca, orthotopic mouse, bioluminescence imaging

## Abstract

**Background:** We investigated whether EphA2 inhibition can attenuate the progression of renal cell carcinoma (RCC) in an orthotopic mouse model of kidney tumor cells (Renca). **Materials and Methods:** 16 BALB/c mice were divided into two groups and implanted with either control or shRNA-mediated, EphA2-knockdown Renca–Luciferase cells via injection under the right renal capsule. Tumor progression was followed by in vivo bioluminescence imaging (BLI). Tumor growth was evaluated via ex vivo BLI and the wet weight of harvested orthotopic kidneys on day 18. Tumor apoptosis was evaluated using the TUNEL assay. Changes in FAK/RhoA signaling, a mediator of malignant cellular behavior, were determined using Western blotting and RT-PCR. **Results:** The TUNEL assay showed increased apoptosis of tumor cells in the EphA2-knockdown group compared to that in the control group (*p* = 0.021). Tumor wet weight (1569.9 ± 595.5 vs. 636.5 ± 288.9 mg, *p* = 0.009) and activation of RhoA and FAK were decreased in the EphA2-knockdown group (*p* < 0.05 for all). Tumor burden was reduced in the EphA2-knockdown group according to in vivo BLI on days 14 and 18 and an ex vivo test (*p* = 0.021, *p* = 0.043, *p* = 0.021). **Conclusions:** EphA2 knockdown significantly reduced the progression of RCC by inducing tumor apoptosis and suppressing FAK/RhoA signaling in an orthotopic mouse model. The EphA2/FAK/RhoA pathway might constitute a potential target to suppress the progression of RCC.

## 1. Introduction

Up to 30–40% of patients with loco-regional renal cell carcinoma (RCC) experience recurrence and progress to advanced distant RCC following radical nephrectomy [1]. However, the effectiveness of conventional treatments, such as chemotherapy, immunotherapy, and radiation therapy, for advanced RCC is currently insufficient [2]. The antiangiogenic effect of inhibiting the vascular endothelial growth factor (VEGF) pathway has been introduced as an effective therapeutic mechanism against RCC [2]. However, the use of VEGFR tyrosine kinase inhibitors did not significantly increase disease-free survival or overall survival in several trials (ATLAS, ASSURE, and PROTECT) [3,4,5]. Therefore, identifying another mechanism to prevent the progression of RCC is necessary to improve the prognosis of RCC.

EphrinA2 (EphA2), a member of the Eph family of receptor tyrosine kinases, is overexpressed in the tumor cells of several types of cancer [6]. EphA2 upregulation may therefore serve as a pathway for reducing the efficacy of tyrosine kinase inhibitors in RCC. Moreover, the EphA2/focal adhesion kinase (FAK)/RhoA signaling pathway has been reported to play an important role in the malignant behavior (i.e., resistance to apoptosis, cell migration, invasion, and angiogenesis) of tumor cells [7,8]. High expression levels of EphA2 are also associated with high-risk RCC and constitute a risk factor for recurrence and poor prognosis following nephrectomy [9].

EphA2 interacts with eight ephrin A-family ligands and progranulin, showing a high preference for ephrin A1 [6]. In normal cells, ephrin A1 on neighboring cells interacts with EphA2 and initiates forward signaling. This forward signaling is mediated via the Ras/MAPK pathway, primarily inhibiting cell proliferation [10]. Conversely, in cancer cells, ephrin A1 is often underexpressed while EphA2 becomes overexpressed. As a result, EphA2 dimerizes with other membrane receptors, such as EGFR and HER2. Moreover, serine 897 (S897) is phosphorylated, which makes EphA2 ligand-independent and activated. This “non-canonical” pathway further activates the Raf/MEK/ERK and Pyk2/Src signaling pathways, contributing to increased cell proliferation and migration. Moreover, activated EphA2 can interact with metalloproteases such as MT1-MMP through Src-dependent signaling, which amplifies downstream oncogenic cascades, including the ERK, PI3K/Akt, and RhoG pathways [11]. Additionally, S897-phosphorylated EphA2 recruits Ephexin4 to activate RhoG and the PI3K/Akt/mTOR signaling pathway, which can promote cell migration, survival, and metastasis [12].

Previous studies have also highlighted the upstream regulatory mechanisms of EphA2 in RCC. In particular, Chen et al. [13] reported that miR-141 suppresses RCC progression by indirectly downregulating EphA2 and inhibiting the FAK/AKT/MMP signaling cascade. However, direct in vivo evidence of EphA2 silencing in an orthotopic and immunocompetent setting has not been reported. Therefore, in the present study, we aimed to directly evaluate the oncogenic role of EphA2 using shRNA-mediated knockdown in a syngeneic RCC model, thereby preserving tumor–immune interactions and enabling longitudinal in vivo monitoring.

Conversely, we previously demonstrated that EphA2 inhibition could attenuate the malignant cellular behavior of RCC cells in vitro [8], similar to previous in vitro results against cellular migration and invasion in non-small-cell lung cancer [14]. Together, these results suggest EphA2 as a potential target for suppressing RCC progression. In this study, we therefore aimed to investigate whether EphA2 depletion could inhibit the progression of RCC in an in vivo orthotopic RCC mouse model.

## 2. Materials and Methods

### 2.1. RCC Cell Line Derivation and Culture

A Renca–luciferase (Luc) stable cell line was established by transfecting mouse kidney tumor (Renca) cells (Korean Cell Line Bank, Seoul, Republic of Korea), commonly used as a orthotopic mouse model for studying RCC [15,16,17], with the FUGW–Luc vector (Molecular Imaging and Neurovascular Research Laboratory, Dongguk University Ilsan Hospital, Goyang, Republic of Korea) as previously described [15]. Prior to transfection, the FUGW–Luc vector was cut with the enzyme XhoI. After transfection, FUGW–Luc-expressing cells were sorted by BD FACSAria II (BD Biosciences, Franklin Lakes, NJ, USA) to select luciferase-expressing Renca–Luc cells. Renca–Luc cells were maintained in RPMI 1640 medium (WELGENE, Gyeongsan, Republic of Korea) containing 10% heat-inactivated fetal bovine serum and 100 μg/mL penicillin–streptomycin.

To prepare shRNA control or EphA2-knockdown RCC stable cell lines, Renca–Luc stable cells were transfected with control or EphA2 short-hairpin RNA (shRNA) lentivirus plasmid vectors (Sigma-Aldrich, St. Louis, MO, USA), as previously described [8,15]. Prior to transfection, 2 × 10^5^ Renca-Luc cells/well were seeded on a 6-well plate and incubated in a humidified atmosphere with 5% CO2 at 37 °C for 24 h. The resultant Renca–Luc cell colonies were incubated for 2 weeks in medium supplemented with 2 μg/mL puromycin, and reverse transcription–quantitative polymerase chain reaction (RT-qPCR) and Western blotting were performed on expanding cultured Renca–Luc EphA2 shRNA- and Renca–Luc control shRNA-transfected cell colonies to evaluate the transfection results.

### 2.2. Reverse Transcription–Quantitative Polymerase Chain Reaction

Total cellular RNA was extracted following the manufacturer’s instructions using the RNeasy Mini Kit (Qiagen, Valencia, CA, USA) and TRIzol reagent (Invitrogen, Carlsbad, CA, USA). Complementary DNA synthesis was performed by aliquoting 1 μg total RNA for each reverse transcription reaction using the RT system (Enzynomics, Daejeon, Republic of Korea) and oligo-deoxythymidine primer. The quantification of RT-qPCR was performed using the EvaGreen qPCR Master Mix Kit (Applied Biological Materials, Richmond, BC, Canada) and LightCycler 480 Real-Time PCR System (Roche, Basel, Switzerland). The results were normalized to β-actin and presented as fold change over control. The primer sequences used for amplification were as follows: 50-GGCTTGTCGGTGTGGTCC-30 as a sense primer and 50-ATGTTGCGGGCAGCCAGGTC-30 as an antisense primer for EphA2 and 50-CCACACCTTCTACAATGAGC-30 as a sense primer and 50-TGAGGTAGTCAGTCAGG TCC-30 as an antisense primer for β-actin.

### 2.3. Western Blot Analysis

Proteins were extracted by lysing the cells (5 × 10^6^) using RIPA buffer supplemented with a protease inhibitor cocktail (Roche, Basel, Switzerland). And because the membrane-bound RhoA represents activated RhoA, we isolated the membrane-bound RhoA using the ProteoExtract Intracellular Proteome Extraction Kit Mini (Calbiochem, San Diego, CA, USA) according to the manufacturer’s instructions. The extracted proteins were loaded separately onto SDS–polyacrylamide gel by size. The separated proteins were then transferred to polyvinylidene difluoride membranes and blocked with 5% skim milk in 0.1% Tween-20 detergent for 1 h. The membranes were incubated overnight at 4 °C with the following primary antibody: anti-EphA2 (1:1000; Santa Cruz Biotechnology, Santa Cruz, CA, USA), anti-FAK (1:2000; Cell Signaling Technology, Danvers, MA, USA), anti-phospho-FAK (Tyr397, 1:1000, Cell Signaling Technology), or anti-RhoA (1:2000, Santa CruzBiotechnology, Dallas, TX, USA). The results were quantified using densitometry and presented as fold change over the control after normalizing to β-actin expression.

### 2.4. Animal Studies

All animal experiments were performed in accordance with Seoul National University Hospital institutional guidelines under the Institutional Animal Care and Use Committee of the Clinical Research Institute (IACUC), an Association for Assessment and Accreditation of Laboratory Animal Care (AAALAC)-accredited facility, at our hospital (approval number: 20-0080; approval date: 22 May 2020). A total of 16 male BALB/c mice (6-week-old) (Central Lab Animal, Seoul, Republic of Korea) were randomly divided into control (*n* = 8) and experimental groups (*n* = 8), in which the Renca–Luc control or Renca–Luc–EphA2 shRNA cell line, respectively, was orthotopically injected into the right kidney. To follow orthotopic tumor engraftment and progression, bioluminescent signals were measured using in vivo bioluminescence imaging (BLI) in half of each group. Mice were euthanized on day 18, and bilateral kidneys were harvested for subsequent analyses: ex vivo BLI, tumor wet weight, TUNEL assay, Western blot, RT-qPCR.

### 2.5. Orthotopic Implantation of Tumor Cells

The implantation of Renca–Luc cells under the renal capsule of BALB/c mice is a well-established orthotopic mouse model for identifying the mechanisms of RCC [16,17]. Renal subcapsular tumor cell implantation was performed following intraperitoneal injection of anesthetics (50 mg/kg Zoletil, 25 mg/kg Rompun). First, the skin under the 13th rib on the dorsal side of the mouse was incised, and the subcutaneous tissue and muscle were dissected using surgical scissors to expose the right kidney. Subsequently, the kidneys were partially retracted by applying mild pressure with a finger nearby. Renca–Luc cells (1 × 10^5^) were inoculated using a Hamilton syringe while moving from the lower pole to the upper pole of the kidney under the renal capsule. Prior to injection, each Renca–Luc cell line was suspended in 50 μL of buffer and then mixed with 50 μL of Matrigel (BD Biosciences). The injection site was pressed for 1 min to prevent recoil of the tumor cell suspension. The muscles and skin were then sutured.

### 2.6. Assessment of RCC Progression and EphA2/FAK/RhoA Pathway Alterations in Orthotopic RCC Kidneys

To compare tumor growth between the two groups of harvested renal tumors, ex vivo BLI and tumor wet weight were evaluated in orthotopic RCC kidneys. Renal tumor wet weight was measured by subtracting the weight of the contralateral normal kidney from the weight of the orthotopic RCC kidney. The mRNA and protein expression of EphA2, FAK, and RhoA were then evaluated by Western blotting and RT-qPCR as previously described for half of each group (*n* = 4) [8]. Activation of FAK and RhoA was assessed by phosphorylation. Prior to representing the results as fold changes over the control group, β-actin was used for normalization.

### 2.7. Detection of Apoptosis in Tumors

Kidneys from the remaining animals in each group (*n* = 4) were subjected to a TUNEL assay to evaluate the extent of apoptosis in the tumors, as previously described [18]. Briefly, orthotopic RCC kidneys were sectioned (5 μm thickness) following deparaffinization and rehydration. In a blind fashion, five high-magnification (×400) fields were randomly selected, and the apoptotic index was evaluated by quantitative image analysis using ImageJ software v.1.53 (National Institutes of Health, Bethesda, MD, USA). The apoptotic index was calculated as the percentage of apoptotic cells relative to the total number of cells in a given area.

### 2.8. In Vivo and Ex Vivo Bioluminescence Imaging

BLI images were obtained 5 min after injecting 150 mg/kg VivoGloTM D-luciferin (Promega, Madison, WI, USA) into the peritoneum under anesthesia. Photons emitted from the tumor during exposure were acquired using a Xenogen IVIS Imaging System 200 (Alameda, CA, USA). The exposure time was set to auto-exposure with a maximum exposure time of 30 s, and the target maximum count (3000 or 30,000 photon counts) was applied. To evaluate the number of cancer cells in the orthotopic RCC kidneys, the number of photons per second (total flux) emitted from the region of interest (ROI) upon in vivo BLI was acquired on days 4, 7, 14, and 18. Ex vivo BLI of the harvested kidneys was performed on day 18. The number of cancer cells was quantified by analyzing the color within the ROI of the image using Living Image v.2.20 (Xenogen).

### 2.9. Statistical Analysis

Data are presented as the mean ± standard deviation. The Mann–Whitney U test was applied to compare continuous variables between the two groups. All statistical tests were two-sided, and *p* < 0.05 was considered statistically significant. Statistical analyses were performed using SPSS software v.27 (IBM Corp., Armonk, NY, USA).

## 3. Results

### 3.1. Generation of an shRNA-Mediated Renca–Luc EphA2-Knockdown Cell Line

To evaluate the effects of EphA2 knockdown in a mouse orthotopic RCC model, we first generated Renca–Luc EphA2-knockdown and control cell lines by transfection with the respective shRNA vectors. Western blot and RT-qPCR (Figure 1) analyses showed that EphA2 protein and mRNA expression, respectively, was significantly decreased in EphA2-knockdown cells compared with that in the control cells (*p* = 0.019, *p* = 0.004).

### 3.2. EphA2 Knockdown Reduces In Vivo Bioluminescence Signal in Orthotopic RCC Mice

Following the injection of EphA2 or control shRNA Renca–Luc cells into the mouse kidney to generate the orthotopic RCC tumor model, luciferase activity in the ROI was determined as the total flux using in vivo BLI (Figure 2A). The total flux gradually increased until day 7 and then rapidly increased thereafter (Figure 2B). The mean total flux was significantly lower in the EphA2-knockdown group than in the control group on day 14 (mean ± standard deviation: 2.71 × 10^9^ ± 0.59 × 10^9^ vs. 3.63 × 10^9^ ± 0.21 × 10^9^, *p* = 0.021) and was further decreased in the EphA2-knockdown group compared to that in the control group by day 18 (4.60 × 10^9^ ± 0.19 × 10^9^ vs. 6.59 × 10^9^ ± 0.16 × 10^9^, *p* = 0.043).

### 3.3. EphA2 Silencing Inhibits RCC Progression

Tumor progression was identified to be more significant in both groups compared to that in the contralateral normal kidney upon ex vivo examination on day 18 (Figure 3A). In the EphA2-knockdown group, tumor progression was reduced compared with that in the control group. The tumor weight in the EphA2-knockdown group was significantly lower than that in the control group (average tumor wet weight: 636.5 ± 288.9 vs. 1569.9 ± 595.5 mg; *p* = 0.009) (Figure 3A). Ex vivo BLI revealed a significant difference in the total flux between the EphA2-knockdown group and control group by day 18 (7.09 × 10^9^ ± 1.88 × 10^9^ vs. 14.9 × 10^9^ ± 1.63 × 10^9^; *p* = 0.021; Figure 3B). No distant metastases were observed in any mouse by day 18 (the end of the experiment) using BLI.

### 3.4. EphA2 Silencing Enhances Apoptosis and Reduces the EphA2/FAK/RhoA Pathway

The results of the TUNEL assay demonstrated that the mean apoptotic index of tumor cells was significantly higher in EphA2-knockdown RCC tissues than in the control RCC tissues on day 18 of tumor maintenance in the orthotopic RCC kidney (7.0 ± 2.3% vs. 1.6 ± 0.9%, *p* = 0.021; Figure 4A).

To determine whether FAK and RhoA are downstream effectors of the EphA2 signaling pathway, EphA2, FAK, and RhoA expression levels were measured using Western blotting and RT-qPCR (Figure 4B). EphA2 protein expression was significantly reduced in the EphA2-knockdown group compared to the control group (*p* = 0.025). In addition, FAK phosphorylation and RhoA phosphorylation levels (as evaluated using the pFAK/FAK and pRhoA/RhoA protein ratios) were also significantly reduced in EphA2-knockdown RCC kidneys compared to those in the control RCC kidneys (*p* = 0.034, *p* = 0.014, respectively). EphA2 mRNA expression was significantly lowered in the EphA2-knockdown group (*p* = 0.001), confirming the stability of the knockdown. Moreover, FAK and RhoA mRNA levels were significantly reduced in EphA2-knockdown RCC kidneys compared to those in the control RCC kidneys (*p* = 0.042, *p* = 0.049, respectively).

## 4. Discussion

In this study, an in vivo mouse model was established using EphA2 shRNA to investigate the effects of EphA2 depletion on malignant cellular behaviors in RCC. The main findings were that the EphA2-knockdown group exhibited (1) decreased numbers of tumor cells, as measured using BLI, (2) increased tumor cell apoptosis, as determined by the TUNEL assay, (3) suppressed FAK/RhoA pathway activation, and (4) reduced tumor wet weight compared to the results in the control group. Together, these findings indicate that EphA2 depletion attenuates RCC progression.

Several studies have described a significant association between EphA2 and RCC [9,19,20,21,22]. In particular, elevated EphA2 expression has been shown to correlate with increased tumor aggressiveness, poor prognosis, and reduced survival in patients with RCC. Talaat et al. compared clinical data and oncological outcomes over a period of 24 months between 50 RCC patients and the publicly available ONCOMINE database [20]. They reported that its overexpression was significantly associated with tumor size, nuclear grade, and tumor stage (all *p* < 0.05) and was also associated with Ki-67 expression, a well-known proliferative marker. Multiple studies have independently confirmed EphA2’s prognostic value, with high expression levels predicting shorter survival times (relative risk, 2.304; 95% CI, 1.102–4.818; *p* = 0.027) [22]. Additionally, EphA2 was also a significant predictive factor of short-term recurrence within 1 year after surgery (vs. 1–5 years recurrence: *p* < 0.001) and decreased overall survival time (*R*^2^ = 0.57, *p* < 0.001) [9].

EphA2 is a membrane protein that regulates cell migration, differentiation, proliferation, morphogenesis, and angiogenesis, playing a major role in cell development and cell-to-cell interaction [10,23,24]. Activation of EphA2 leads to decreased extracellular matrix (ECM) attachment and induction of intracellular signaling networks. These actions of EphA2, which are involved in cell adhesion/repulsion or regulate cell survival/death, are mediated through various pathways, such as the FAK, extracellular-regulated protein kinase (ERK), Akt, Src family kinase, Ras/MAPK, or PI3K/Akt pathways. However, overexpression of EphA2 is observed in several solid cancers, such as prostate, ovarian, breast, cervical, colorectal, and lung cancers, including RCC [6]. These outcomes are related to malignant cellular behaviors, such as apoptosis resistance and cellular invasiveness, as a result of EphA2 overexpression [8,14]. Alternatively, our orthotopic mouse study showed that EphA2 knockdown reduced the apoptosis resistance of tumor cells, decreased the overall tumor cell number, and lowered the tumor weight, further confirming the cancer-associated functions of EphA2.

Several prior studies have addressed the mechanism by which EphA2 affects malignant cellular behavior. In our previous in vitro study, we reported that the malignant cellular behaviors of RCC cells induced by activation of the FAK/RhoA signaling pathway were attenuated via EphA2 inhibition [8]. In turn, Lee et al. reported that the suppression of non-small-cell lung cancer invasion and metastasis by inhibiting EphA2 was associated with a decrease in FAK expression on Western blots [14]. EphA2 inhibition slowed the healing rate of wounded cultures and reduced the number, density, and size of tumor colonies. Woo et al. reported on the mechanism by which EphA2 induces cancer progression [25]. The *EPHB6* mutation had an oncogenic effect by inhibiting degradation of EphA2 in cells, which activated JNK/CDH11/RhoA/FAK downstream signaling. Activated FAK and RhoA are also known to promote cell migration, invasion, apoptosis suppression, and angiogenesis in various cancers [26,27].

In addition, Chen et al. [13] demonstrated that miR-141 suppresses RCC progression by downregulating EphA2 and inhibiting the FAK/AKT/MMP signaling cascade. Unlike that study, we directly silenced EphA2 using shRNA in a syngeneic, immunocompetent orthotopic model, providing in vivo mechanistic evidence that EphA2 depletion attenuates tumor growth and apoptosis resistance while suppressing the EphA2/FAK/RhoA axis.

Beyond its oncogenic signaling role, EphA2 has also been implicated in therapeutic resistance, underscoring its clinical relevance in RCC management. In metastatic RCC, overexpression of EphA2 and Y-box binding protein 1 (YB1) was found to mediate resistance to the VEGFR-TKI sunitinib, whereas pharmacologic inhibition of EphA2 with ALW-II-41-27 restored sunitinib sensitivity both in vitro and in vivo [21]. In addition, EphA2 overexpression has been associated with resistance to multiple EGFR-targeted TKIs in other malignancies, where EphA2 blockade with ALW-II-41-27 reversed acquired resistance and re-sensitized tumor cells to erlotinib and afatinib [10]. Collectively, these findings indicate that EphA2 not only promotes tumor aggressiveness but also contributes to adaptive resistance against tyrosine kinase inhibitors. Therefore, integrating EphA2 inhibition with existing VEGFR-TKIs may provide a rational strategy to overcome therapeutic resistance and enhance treatment outcomes in RCC.

BLI is based on the bioluminescence, where living organisms emit light due to the reaction between the molecule luciferin and oxygen, catalyzed by the enzyme luciferase [28,29]. This imaging test is performed by injecting luciferase into the target cells, and a sensitive detector is used to capture the total flux of emitted light. Therefore, BLI is primarily employed to observe physiological processes and biological activities occurring in real time within living systems. In cancer research using animal models, BLI is used as an in vivo imaging test that quickly and sensitively identifies increased tumor burden or metastases without requiring euthanization of the experimental animals [28,30,31]. In the present study, the results of in vivo BLI analyses revealed that the total flux in the EphA2-knockdown group was reduced on days 14 and 18 compared to the control group (*p* = 0.021, *p* = 0.043, respectively). In addition, ex vivo BLI demonstrated a significant relative decrease (*p* = 0.021) compared to the control group. We also obtained histological evidence that EphA2 inhibition suppressed RCC progression, as indicated by the enhanced tumor cell apoptosis revealed using the TUNEL assay.

The BLI test in the orthotopic RCC mouse model should be conducted and interpreted meticulously. There are several factors known to interfere with the correlation between BLI signal and tumor burden. Larger, progressed tumors may not express proportional light signals because the ATP or oxygen supplied to the luciferase may be blocked owing to tissue necrosis or hypoxia [32]. Consistent with this, the correlation between BLI results and tumor burden was reported to be low, especially for large tumors (>1.2 cm) [33]. Another limitation of BLI is that it is relatively inaccurate for hemorrhagic lesions or lesions with increased blood flow owing to the reduced passage of light through the hemoglobin of red blood cells [34,35]. Fluids such as ascites can also lower BLI signal intensity [36]. The location of the target organ may be a confounding factor because the signal intensity of BLI decreases ten-fold for every 1 cm increase in depth (26). In particular, the pathological features of RCC, such as its hypervascularity and coagulative tumor necrosis, could reduce the signal accuracy of BLI [37].

There are also several limitations. In additional research, conducting studies with a larger number of experimental subjects would lead to more solid conclusions. Furthermore, the study should be expanded to an orthotopic model using human RCC cells instead of murine RCC cells for a more clinical impression. We can also consider alternative testing methods to follow the orthotopic RCC model with in vivo longitudinal monitoring, such as fluorescence imaging, which is not affected by the metabolism of cancer cells. Nevertheless, this study demonstrates the potential for inhibition of EphA2 in RCC to suppress tumor progression.

## 5. Conclusions

This study provides in vivo evidence identifying EphA2 as a potential target to mediate the suppression of tumor progression in RCC. Using an orthotopic and immunocompetent mouse model, EphA2 knockdown resulted in significantly reduced tumor wet weight and increased tumor cell apoptosis compared to those in the control group in the orthotopic RCC mouse model. In this process, reduced FAK/RhoA signaling, a downstream effector of EphA2, was associated with attenuated tumor progression in RCC. Further studies are needed to obtain clearer evidence that EphA2 is a potential therapeutic target for suppressing the progression of RCC.

## Figures and Tables

**Figure 1 cells-14-01981-f001:**
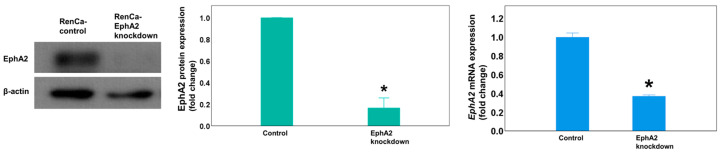
In vitro transfection with the EphA2 shRNA plasmid virus depletes EphA2 expression in Renca cells. Representative Western blot showing EphA2 protein expression in transfected colony with EphA2 or control shRNA plasmid virus. *EphA2* gene expression as measured by RT-qPCR. The results were normalized to those of β-actin and are presented as fold changes. * *p* = 0.019 and *p* = 0.004, respectively. EphA2 = EphrinA2; RT-qPCR = real-time quantitative reverse transcription–polymerase chain reaction.

**Figure 2 cells-14-01981-f002:**
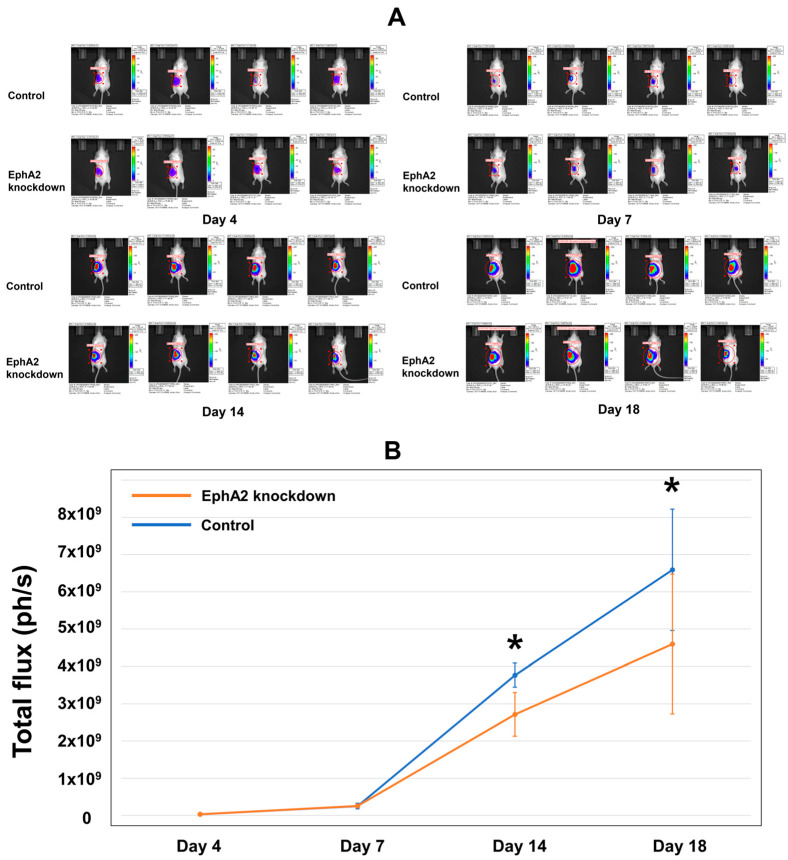
In vivo BLI evaluation reveals reduced tumor progression after establishment of the orthotopic RCC kidney model using EphA2-knockdown cells. BLI analysis revealed that orthotopic RCC cells were well engrafted, and the tumor burden increased over time. (**A**) Images showing the in vivo tumor burden, as indicated by bioluminescence on days 4, 7, 14, and 18. (**B**) Quantitation of the mean total flux over time for the two groups (photons per second, ph/s). * Day 14: *p* = 0.021; day 18: *p* = 0.043. BLI = bioluminescence imaging; RCC = renal cell carcinoma; EphA2 = EphrinA2; RT-qPCR = real-time quantitative reverse transcription–polymerase chain reaction.

**Figure 3 cells-14-01981-f003:**
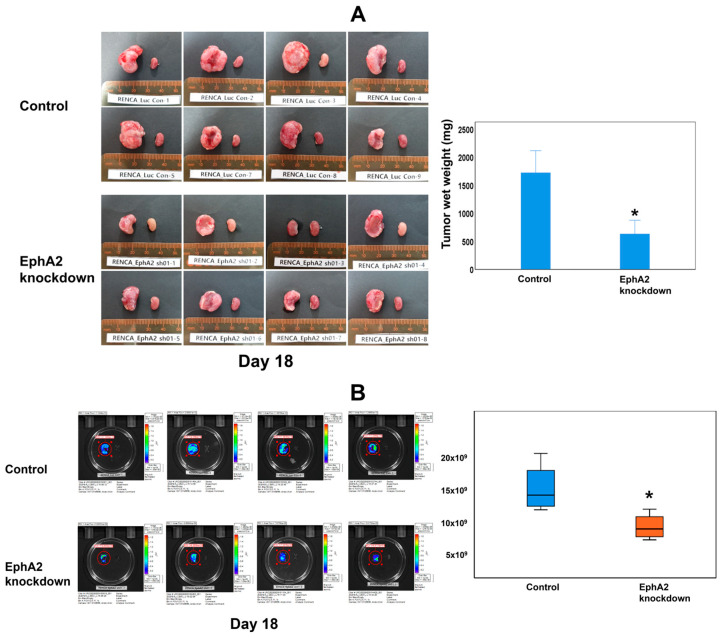
EphA2 knockdown attenuates tumor progression, as shown by ex vivo evaluation of excised kidneys after euthanasia of mice on day 18. (**A**) Gross progression of the tumor; bilateral kidneys of each mouse and the relative tumor wet weight for each group (* *p* = 0.009). (**B**) Ex vivo bioluminescence images and quantitation of the mean total flux of the two groups (* *p* = 0.021) showed significantly reduced tumor growth in EphA2-knockdown orthotopic RCC kidneys. EphA2 = EphrinA2; RCC = renal cell carcinoma.

**Figure 4 cells-14-01981-f004:**
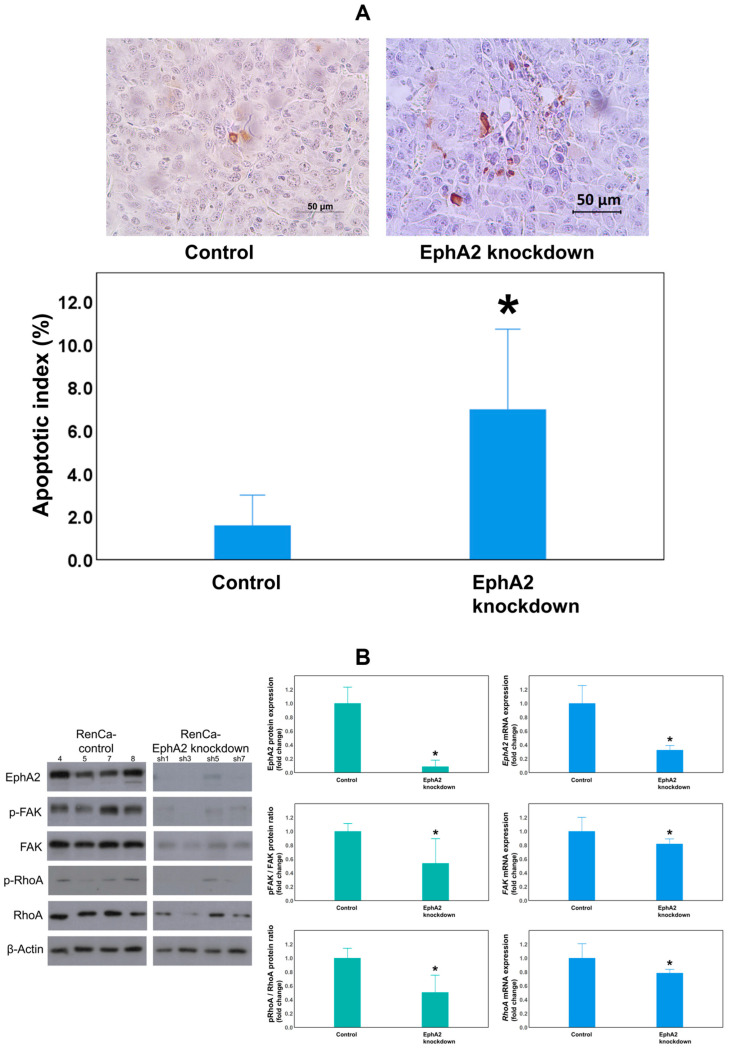
The EphA2-knockdown group showed increased apoptosis of tumor cells and decreased EphA2/FAK/RhoA signaling pathway expression. (**A**) Representative images from TUNEL staining (top) and quantitation of the tumor cell apoptotic index in each group (* *p* = 0.021). The EphA2/FAK/RhoA signaling pathway was evaluated by (**B**) Western blot analysis of EphA2, pFAK/FAK, and pRhoA/RhoA expression in protein lysates obtained from orthotopic RCC tumors of the control (RenCa-control) and EphA2-knockdown (RenCa–EphA2 shRNA) groups (β-actin as a loading control). RT-qPCR analysis of EphA2, FAK, and RhoA mRNA expression in corresponding tumor tissues (β-actin as an internal control). Quantification of the results is shown. * (**B**): *p* = 0.025 (Epha2 protein), *p* = 0.034 (phosphorylated FAK (pFAK)/FAK protein), *p* = 0.014 (phosphorylated RhoA (pRhoA)/RhoA protein); *p* = 0.001 (Epha2 mRNA), *p* = 0.042 (FAK mRNA), *p* = 0.049 (RhoA mRNA). EphA2 = EphrinA2; FAK = focal adhesion kinase; RhoA = Ras homolog family member A; RT-qPCR = real-time quantitative reverse transcription–polymerase chain reaction.

## Data Availability

The dataset for the current study is available from the corresponding author on reasonable request.

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
