# Peer review of "Effect of EphA2 Silencing on Inhibiting the Progression of Renal Cell Carcinoma in an Orthotopic Mouse Model"

_cells, 2025, doi:10.3390/cells14241981_

Round 1
Reviewer 1 Report (Previous Reviewer 2)
Comments and Suggestions for Authors
The authors have convincingly addressed most of my previous comments.
The manuscript has been improved.
For clarity, authors need to label figures at the top of each figure's panel.
Author Response
Comments 1: For clarity, authors need to label figures at the top of each figure's panel.
Response 1: We appreciate the reviewer’s helpful suggestion.
Accordingly, we have revised all figures to clearly include panel labels (A, B, C, etc.) at the top of each panel. These changes improve the readability and clarity of the figure layout.
All updated figures have been incorporated into the revised manuscript.
Reviewer 2 Report (Previous Reviewer 1)
Comments and Suggestions for Authors
The authors have addressed my previous comments. I recommend including small-molecule EphA2 inhibitors to complement the data and confirm on-target effects.
Author Response
Comments 1: I recommend including small-molecule EphA2 inhibitors to complement the data and confirm on-target effects.
Response 1: We sincerely appreciate the reviewer’s thoughtful recommendation. We agree that evaluating small-molecule EphA2 inhibitors would further strengthen the evidence for on-target effects. However, we currently do not have any remaining stored tumor or protein samples suitable for additional inhibitor-based assays, and generating new samples would require reconstructing the entire orthotopic model, which is not feasible during the revision process. Although we were unable to include these data at this stage, we fully recognize their importance and plan to incorporate such analyses in future studies.
This manuscript is a resubmission of an earlier submission. The following is a list of the peer review reports and author responses from that submission.
Round 1
Reviewer 1 Report
Comments and Suggestions for Authors
This study explored whether depleting EphA2 can impede RCC advancement in a mouse model. By validating EphA2 knockdown through protein and mRNA analyses, the study employs in vivo bioluminescence imaging (BLI) to exhibit reduced tumor growth. Ex vivo analyses further unveiled lower tumor weights and heightened tumor cell apoptosis in the EphA2-depleted group. Additionally, the study highlighted suppressed activation of the EphA2/FAK/RhoA pathway, suggesting EphA2's involvement in RCC progression. In summary, EphA2 knockdown reduced tumor progression and elevated apoptosis in the RCC mouse model, implying EphA2's promise as a therapeutic target. However, enhancing the manuscript could involve:
1) The manuscript would benefit from a discussion on the potential clinical implications of targeting EphA2 in RCC therapy, highlighting its relevance to current treatments and therapeutic prospects.
2) In addition to the EphA2 knockdown in vivo experiment, incorporating pharmacological inhibitors of EphA2 could offer further validation of its influence on tumor growth and signaling pathways.
3) To enhance the study's comprehensiveness, conducting transcriptomic analyses such as bulk RNA sequencing would be valuable in identifying gene expression alterations resulting from EphA2 depletion and validating downstream effects.
Minor revisions:
Line 10: 'reanl' to 'renal'.
Line 82-85: '50' and '30' corrected to '5' and '3'.
Author Response
Point-by-point response to Reviewer 1
Comments 1: The manuscript would benefit from a discussion on the potential clinical implications of targeting EphA2 in RCC therapy, highlighting its relevance to current treatments and therapeutic prospects.
Response 1: Thank you for this insightful comment. In response, we have revised the Discussion section (page 11, line 316) by replacing the sentences beginning with “Another RCC study reported …” with a new paragraph that specifically addresses the clinical relevance of EphA2 inhibition and its relationship to current RCC therapies, as follows:
Beyond its oncogenic signaling role, EphA2 has also been implicated in therapeutic resistance, underscoring its clinical relevance in RCC management. In metastatic RCC, overexpression of EphA2 and Y-box binding protein 1 (YB1) was found to mediate resistance to the VEGFR-TKI sunitinib, whereas pharmacologic inhibition of EphA2 with ALW-II-41-27 restored sunitinib sensitivity both in vitro and in vivo [17]. In addition, EphA2 overexpression has been associated with resistance to multiple EGFR-targeted TKIs in other malignancies, where EphA2 blockade with ALW-II-41-27 reversed acquired resistance and re-sensitized tumor cells to erlotinib and afatinib [21]. Collectively, these findings indicate that EphA2 not only promotes tumor aggressiveness but also contributes to adaptive resistance against tyrosine kinase inhibitors. Therefore, integrating EphA2 inhibition with existing VEGFR-TKIs may provide a rational strategy to overcome therapeutic resistance and enhance treatment outcomes in RCC.
We believe this revision adequately addresses the reviewer’s concern by clarifying the clinical and translational implications of EphA2 targeting in the context of current RCC treatment strategies.
Comments 2 & 3
In addition to the EphA2 knockdown in vivo experiment, incorporating pharmacological inhibitors of EphA2 could offer further validation of its influence on tumor growth and signaling pathways.
To enhance the study's comprehensiveness, conducting transcriptomic analyses such as bulk RNA sequencing would be valuable in identifying gene expression alterations resulting from EphA2 depletion and validating downstream effects.
Response 2 & 3: We sincerely appreciate these thoughtful and constructive suggestions. We fully agree that additional experiments, such as pharmacological inhibition of EphA2 and transcriptomic analyses including RNA sequencing, would further strengthen the mechanistic depth and comprehensiveness of our study. However, due to the limited availability of tumor tissue and experimental resources, performing additional in vivo or transcriptomic analyses is not feasible at this stage. We recognize the importance of these approaches and plan to incorporate both EphA2 inhibitor–based validation and RNA sequencing analyses in our future studies to expand the understanding of EphA2-mediated molecular mechanisms in RCC.
Reviewer 2 Report
Comments and Suggestions for Authors
In this work Lee et al. have investigated whether the deficiency of EphA2, an important predictive biomarker of renal cell carcinoma (RCC) that is associated with poor disease outcome, may attenuate growth tumour in RCC. To study this issue, the authors transduced mouse kidney tumour cells with EphA2-specific shRNA lentivirus vectors and transferred EphA2-deficient cells to BALB/c mice. Using this orthotopic mouse model of kidney carcinoma, the authors performed a comparative study of RCC growth progression, apoptosis and EphA2-mediated signalling between control and EphA2-deficient kidney carcinoma cells.
Major comments
The work by Lee et al. represents an interesting in vivo approach to confirm the function of EphA2 in RCC progression, providing some evidence of the potential role of EphA2 as a pharmacological target for the treatment of RCC. However, it is worth noting that a closely related previous study by Chen et al., 2014 (miR-141 Is a Key Regulator of Renal Cell Carcinoma Proliferation and Metastasis by Controlling EphA2 Expression. Clin Cancer Res; 20(10); 2617–30.) has already demonstrated a role for the interference/deficiency of EphA2 in reducing tumour cell growth and that this role can take place through the inhibition of EphA2 signalling. In the quoted study, the authors identified the microRNA miR-141 as a negative regulator of EphA2 expression in kidney cancer cells and showed that RCC patients express lower amounts of miR-141 in the kidney cancer cells than in control kidney tissue. Moreover, the mentioned previous work also studied cell growth and EphA2 signalling in mouse kidney cancer cells transfected with siRNAs specific for EphA2 and observed that, in vitro, EphA2-deficient cells were defective in cell growth and FAK phosphorylation as Lee et al. observed in vivo. In view of this previous solid publication and its interesting pathophysiological insights, the manuscript by Lee et al. could be considered of limited novelty and interest. In fact, only two points remain interesting for this reviewer: i) EphA2 deficiency generated by EphA2-specific shRNA lentivirus vectors seems a specific method to delete EphA2; and ii) The studies by Lee et al. with EphA2-deficient kidney carcinoma cells have been exclusively performed in vivo, which make their results relevant.
Some specific points
1.- The ‘Introduction’ section should be extended and considerably improved. More detailed information about Ephrin receptors and their ligands, signalling pathways (src kinases, PI3K-Akt, metalloproteinase…), expression and functions in normal and tumour tissues should be included.
2.- The manuscript by Chen et al., 2014 should be properly quoted and discussed.
3.- Figures are in general poor. They must be improved and better organized (e.g. graphs are sometimes oversized).
4.- Figure 4 B is not convincing at all. What sample has been loaded in each Western blot’s lane? Why the quantified ratios of pRhoA/RhoA and pFAK/FAK are so similar being total FAK and RhoA WBs so different? Results from this figure are key, and the authors must considerably improve/repeat this set of in vivo experiments, including additional analysis of pAkt, Akt, and MMP-9 expression in cancer cells from the kidney as important signal transduction pathways that can be triggered from EphA2.
5.- ‘Conclusions’ section, page 11, line 337: The strength of the first statement should be reduced. Chen et al. have already proposed EphA2 as a ‘potential’ target to mediate tumour suppression in RCC.
Author Response
Point-by-point response to Reviewer 2
Comments 1: The ‘Introduction’ section should be extended and considerably improved. More detailed information about Ephrin receptors and their ligands, signalling pathways (src kinases, PI3K-Akt, metalloproteinase…), expression and functions in normal and tumour tissues should be included.
Response 1: Thank you for your valuable suggestion. We have revised the Introduction section to include a more detailed explanation of EphA2-related signaling mechanisms and ligand interactions. Specifically, we have added the following paragraph on page 1, line 58 of the revised manuscript:
EphA2 interacts with eight ephrin A-family ligands and progranulin, showing a high preference for ephrin A1 [6]. In normal cells, ephrin A1 on neighboring cells interacts with EphA2 and initiates forward signaling. This forward signaling is mediated via the Ras/MAPK pathway, primarily inhibiting cell proliferation [10]. Conversely, in cancer cells, ephrin A1 is often underexpressed while EphA2 becomes overexpressed. As a result, EphA2 dimerizes with other membrane receptors, such as EGFR and HER2. Moreover, serine 897 (S897) is phosphorylated, which makes EphA2 ligand-independent and activated. This “non-canonical” pathway further activates the Raf/MEK/ERK and Pyk2/Src signaling pathways, contributing to increased cell proliferation and migration. Moreover, activated EphA2 can engage metalloproteases such as MT1-MMP through Src-dependent signaling, which amplifies downstream oncogenic cascades, including ERK, PI3K/Akt, and RhoG pathways [11]. Additionally, S897-phosphorylated EphA2 recruits Ephexin4 to activate RhoG and the PI3K/Akt/mTOR signaling pathway, which can promote cell migration, survival, and metastasis [12].
The newly added references are as follows:
[11] Toracchio, L.; Carrabotta, M.; Mancarella, C.; Morrione, A.; Scotlandi, K. EphA2 in Cancer: Molecular Complexity and Therapeutic Opportunities. Int. J. Mol. Sci. 2024, 25(22), 12191.
[12] Murga, C.; Laguinge, L.; Wetzker, R.; Cuadrado, A.; Gutkind, J.S. Rac1 and RhoG Promote Cell Survival by the Activation of PI3K and Akt, Independently of Their Ability to Stimulate JNK and NF-κB. Oncogene 2002, 21(2), 207–216.
We believe this revision provides a more comprehensive overview of the EphA2 signaling network and better addresses the reviewer’s concern.
Comments 2: The manuscript by Chen et al., 2014 should be properly quoted and discussed.
Response 2: We appreciate this important point and have now properly cited and discussed Chen et al. (Clin Cancer Res, 2014). While Chen et al. showed that miR-141 suppresses RCC progression by indirectly downregulating EphA2 and inhibiting FAK/AKT/MMP signaling, our study directly silences EphA2 using shRNA in a syngeneic orthotopic and immunocompetent model, thereby preserving the tumor immune microenvironment.
We further provide in vivo evidence that EphA2 depletion reduces longitudinal tumor burden (BLI), tumor wet weight, and apoptosis resistance (TUNEL), while attenuating the EphA2/FAK/RhoA axis.
Importantly, by integrating current literature, we also position EphA2 within therapeutic-resistance contexts to underscore its translational relevance beyond the miRNA regulatory framework.
We have added a new paragraph to the Introduction and to the Discussion section as follows:
Introduction (page 1, line 58): Previous studies have also highlighted the upstream regulatory mechanisms of EphA2 in RCC. In particular, Chen et al. [13] reported that miR-141 suppresses RCC progression by indirectly downregulating EphA2 and inhibiting the FAK/AKT/MMP signaling cascade. However, direct in vivo evidence of EphA2 silencing in an orthotopic and immunocompetent setting has not been reported. Therefore, in the present study, we aimed to directly evaluate the oncogenic role of EphA2 using shRNA-mediated knockdown in a syngeneic RCC model, thereby preserving tumor–immune interactions and enabling longitudinal in vivo monitoring.
Discussion (page 11, lines 316-323): In addition, Chen et al. [13] demonstrated that miR-141 suppresses RCC progression by downregulating EphA2 and inhibiting the FAK/AKT/MMP signaling cascade. Unlike that study, we directly silenced EphA2 using shRNA in a syngeneic, immunocompetent orthotopic model, providing in vivo mechanistic evidence that EphA2 depletion attenuates tumor growth and apoptosis resistance while suppressing the EphA2/FAK/RhoA axis.
Comment 3: Figures are in general poor. They must be improved and better organized (e.g., graphs are sometimes oversized).
Response 3: We appreciate the reviewer’s constructive feedback regarding the overall quality and organization of the figures. In the revised manuscript, some figures have been reformatted to improve visual clarity, consistency, and proportionality across panels (Figures 1a, 3a, 4b).
Comment 4: Figure 4B is not convincing at all. What sample has been loaded in each Western blot’s lane? Why the quantified ratios of pRhoA/RhoA and pFAK/FAK are so similar being total FAK and RhoA WBs so different? Results from this figure are key, and the authors must considerably improve/repeat this set of in vivo experiments, including additional analysis of pAkt, Akt, and MMP-9 expression in cancer cells from the kidney as important signal transduction pathways that can be triggered from EphA2.
Response 4: To address the reviewer’s concern regarding the sample identity, Figure 4B has been modified to explicitly state that each lane represents protein lysates derived from orthotopic renal tumors of the control (RenCa-control) and EphA2-knockdown (RenCa–EphA2 shRNA) groups(4, 5, 7 and 8 for the control group; sh1, sh3, sh5, and sh7 for the EphA2-knockdown group).
To enhance the visual clarity and consistency of the Western blot data, the contrast and brightness of all panels were uniformly adjusted under identical exposure settings, ensuring that the band intensities accurately represent the original signal levels. These adjustments improve visibility while maintaining the integrity of the original data.
In addition, all Western blot results were re-quantified and the bar graphs were regenerated based on the fold change over the control group after normalization to β-actin expression. Because these quantitative values represent normalized fold-change calculations rather than raw band intensities, the resulting bar graphs may not correspond to each Western blot band in a one-to-one manner, but they accurately reflect the relative signal levels after normalization.
Regarding the reviewer’s request for additional pAkt, Akt, and MMP-9 analyses, we fully acknowledge the importance of these downstream pathways. However, because the remaining tumor tissue and experimental resources are limited, additional in vivo or transcriptomic analyses cannot be performed at this stage.
The revised Figure 4 legend now reads as follows:
Figure 4. The EphA2-knockdown group showed increased apoptosis of tumor cells and decreased EphA2/FAK/RhoA signaling pathway expression in vivo.
(B) Western blot analysis of EphA2, pFAK/FAK, and pRhoA/RhoA expression in protein lysates obtained from orthotopic RCC tumors of the control (RenCa-control) and EphA2-knockdown (RenCa–EphA2 shRNA) groups (β-actin as a loading control).
Comment 5: 'Conclusions’ section, page 11, line 337: The strength of the first statement should be reduced. Chen et al. have already proposed EphA2 as a “potential” target to mediate tumour suppression in RCC.
Response 5: We thank the reviewer for this valuable comment. We agree that the original statement may have overstated the novelty of our findings, as Chen et al. (Clin Cancer Res, 2014) previously demonstrated that EphA2 could mediate tumor suppression through miR-141 regulation in RCC. To address this point, we have revised the conclusion to moderate the claim and clarify our specific in vivo contribution. In addition, Chen et al.’s work has now been properly cited and discussed in both the Introduction and Discussion sections to provide appropriate context and distinction from our study. The revised text now highlights that our study provides in vivo evidence supporting the therapeutic potential of EphA2 using an orthotopic and immunocompetent mouse model, rather than claiming to be the first to identify EphA2 as a target.
Revised Text (page 11, line 337):
This study provides in vivo evidence identifying EphA2 as a potential target to mediate the suppression of tumor progression in RCC. Using an orthotopic and immunocompetent mouse model, EphA2 knockdown resulted in significantly reduced tumor wet weight and increased tumor cell apoptosis compared with the control group. In this process, reduced FAK/RhoA signaling, a downstream effector of EphA2, was associated with attenuated tumor progression in RCC. Further studies are needed to obtain clearer evidence that EphA2 is a potential therapeutic target for suppressing RCC progression.